# The Disparity in Mental Health Between Two Generations of Internal Migrants (IMs) in China: Evidence from A Nationwide Cross-Sectional Study

**DOI:** 10.3390/ijerph16142608

**Published:** 2019-07-22

**Authors:** Wen Chen, Qi Zhang, Andre M.N. Renzaho, Li Ling

**Affiliations:** 1Department of Medical Statistics, School of Public Health, Sun Yat-sen University, #74 Zhongshan Road 2, Guangzhou 510080, China; 2Center for Migrant Health Policy, Sun Yat-sen University, #74 Zhongshan Road 2, Guangzhou 510080, China; 3School of Social Science and Psychology, Western Sydney University, Locked Bag 1797, Penrith, NSW 2751, Australia; 4School of Community and Environmental Health, Old Dominion University, Room 3138, Health Sciences Building, Norfolk, VA 23529, USA

**Keywords:** mental health, migrant, disparity, generation

## Abstract

Background: Internal migrants (IMs) are a large, vulnerable population in China and are mostly driven by national economic reform. IMs who were born before and after 1980, when the general social and economic reform began to appear in China, are considered to be two separate generations. The generational differences in mental health across IMs remain undocumented. In this study, the intergenerational disparity in IMs’ mental health, using data from a national cross-sectional study, was assessed. Methods: Cross-sectional data from the “National Internal Migrant Dynamic Monitoring Survey 2014” were used. IMs were divided into the “old” or “new” generation, based on their date of birth (before 1980 vs. from 1980 onwards). Mental health includes psychological distress, which was measured using the Kessler Screening Scale for Psychological Distress (K6), and perceived stress, which was measured with the Perceived Stress Scales (PSS-4). Two-level Generalized Linear Mixed Models were performed so as to assess the generation gap and associated factors of each group’s mental health. IM demographics, migration characteristics, and social integration indicators were controlled for when assessing the intergenerational disparity in mental health. Results: A total of 15,999 IMs from eight different cities participated in the survey. New generation migrants accounted for 61.5% (9838/15,999) of the total sample. After controlling for participants’ characteristics, new generation migrants had higher psychological distress scores (*β_ad_* = 0.084, 95% CI: (0.026,0.193) and higher perceived stress scores (*β_ad_* = 0.118, 95% CI: 0.029, 0.207) than the older generation. For both generations, factors associated with good mental health included high levels of social integration, personal autonomy, and life satisfaction, as well as self-rated good physical health. For the new generation, the mental health of urban-to-urban IMs (*β*_ad_ = 0.201, 95%CI: 0.009, 0.410) for the K6, *β*_ad_ = 0.241, 95% CI: 0.073, 0.409 for the PSS-4), IMs with a longer migration duration (*β*_ad_ = 0.002, 95% CI: (0.000, 0.003) for the PSS-4) and IMs with a higher annual income (*β*_ad_ = 0.124, 95% CI: (0.029, 0.218) for the K6) was significantly poorer than their counterparts. Conclusions: New-generation migrants’ mental health is worse compared to older IMs. An array of services for addressing these generation-specific needs may facilitate the promotion of mental health among IMs in China.

## 1. Introduction

By the end of 2014, there were 253 million internal migrants (IMs) in China, who accounted for nearly one-fifth of China’s overall population and one-third of IMs in the world [1,2]. The emergence of IMs in China can be linked to the *Hukou* (in Chinese) system, a household registration management system, which was introduced in the late 1950s. Under this system, districts (in urban areas) and counties (in rural areas) were defined as the lowest management division of the *Hukou* system, which meant that people were restricted to voluntarily moving out of their registered district/county for a permanent residence purpose. However, since the 1980s, China has instigated a market-oriented economic reform to boost the economy. The Chinese government has encouraged the development of urban enterprises, and a greater amount of foreign investment has begun to flow into the eastern and southern coastal regions of China. Due to high demand for labour in urban areas, the Chinese government has loosened the reigns of control regarding *Hukou* management, allowing agricultural *Houkou* populations to work and do business in urban areas. As a result of this change, many people have left their registered place of residence and have now begun to move to relatively developed areas for a better life [3,4]. In China, this migrant population has been defined as IMs. Consistent with the *Hukou*’s specifications, IMs are classified as being temporary residents in host cities.

### 1.1. Migration and Health

Previous research findings on migration and health among international migrants have postulated the “healthy migrant paradox”, which stipulates that recent migrants from developing countries appear to have better overall health, especially physical health, than permanent residents in host countries and those who remain behind in their own countries; unfortunately this health advantage dissipates over time [5,6,7,8]. It has been hypothesised that migration itself is physically demanding and migrants tend to self-select jobs that have a high physical burden [9,10]. The pre-migration self-selection process and health screening, and cultural buffering and employability of host countries cannot solely explain this epidemiological paradox. The fact that, migrants have a lower socioeconomic status and pooer access to healthcare than the host population [7,8], there are other factors that could explain the healthy migrant effect. A common explanation for this is the “salmon bias hypothesis” [11,12], which describes the voluntary and selective return-migration of the elderly and the sick, as well as those who are weak and are unable to physically and mentally adapt to their new working and living envirnment. Most countries and regions do not have adequate surveillance systems to catpture the demographic and health characteristics of returnee migrants. Therefore, this under-reporting of health conditions among migrants who remain put in host societies creates an artificially healthy migrant phenomenon.

Recently, the healthy migrant effect has also been described amongst IMs in China [13,14]. However, the healthy migrant effect is not clear in terms of mental health. Paradoxically, preventive mental health interventions and services remain unevenly implemented across China [13,14]. Mental health among IMs in China is still a prominent public health issue and results from existing studies are mixed. Some studies indicate that mental health poorer among IMs than residents in receiving and sending regions [15,16]. Other studies have reported that mental health amongst IMs is comparable to or even better than residents in receiving and sending regions [13,17,18]. It is possible that the pre-migration self-selection process may explain this mental health advantage among IMs. Another school of thought asserts that IMs who experience migration develop some form of stress, which can arise from family separation [19], difficulties with social integration [14], and stigma and discrimination by registered residents [20,21]. In addition, they are more vulnerable to suboptimal working and living circumstances than the receiving societies. This includes taking on physically demanding jobs, working for long hours, and being largely excluded from accessing welfare services available to host residents, such as health care, insurance, and superannuation [3,4,22].

### 1.2. The Intergenerational Disparity in Migrants’ Mental Health

In accordance with the increasing number of migrants globally [23], the intergenerational disparity in migrants’ mental health is a meaningful research topic. Within immigration studies, the term “generation” is often used to refer to immigrants who are born in their country of origin (i.e., the first generation) or destination (i.e., the second generation). First-generation immigrants face numerous social integration challenges, such as adapting to the socioculture aspects, language, and customs of receiving communities, much more so than the second generation. Previous studies have examined intergenerational differences in mental health amongst immigrants, all of which have produced mixed results. For example, similar rates of common mental disorders in first- and second-generation immigrants were found in the United States and Israel [24,25]. However, evidence obtained from Germany indicates better mental health in second-generation immigrants than in first-generation immigrants [26]. Moreover, recent research in the United Kingdom found that differences in immigrants’ mental health varied across generations and ethnic groups. For most ethnic groups, first-generation immigrants who had lived in the United Kingdom for less than ten years had the highest levels of mental health, followed by second-generation and first-generation immigrants who had lived in the UK for ten years or more [27]. Existing studies suggest that acculturative stress is related to changes in one’s health behaviour and lifestyle, as well as that the intergenerational acculturation gap (the differing pace of acculturation between children and parents) is associated with the intergenerational disparity in immigrants’ mental health [28,29].

For IMs, social integration challenges may be not as remarkable as those among immigrants. Therefore, the term “generation” has a different meaning for IMs. Defined by the government, “new-generation” migrants are individuals who were born from 1980 onwards when China first began its economic reform and adopted the “one-child” policy [30,31]. In 2013, the new generation accounted for 46.6% of IMs in China [30]. Influenced by the ever changing social and family structure, which is a result of the economic reform and “one-child” policy, these young migrants have a very different socio-demographic profile compared to previous generations (who are referred to as “old-generation” migrants), including inter alia: having a higher female population, a higher level of educational attainment, higher career aspirations, higher preference for urban lifestyles, more diverse social networks, and increased struggles with self-identity [30,32]. Current evidence suggests that these characteristics may lead to inequalities in mental health between generations, but there is little empirical evidence to support this claim [33,34,35]. This study aims to fill in this knowledge gap, to assess the intergenerational disparities in mental health amongst IMs in China and to explore correlative factors of mental health among different migrant generations.

## 2. Methods

### 2.1. Data Resource

The current study used data from the “National Internal Migrant Dynamic Monitoring Survey, 2014”, which is a national cross-sectional study funded and conducted by the National Population and Family Planning Commission of China in May 2014.

### 2.2. Study Participants and Data Collection

The study population included 18- to 59-year-old IMs who had lived in the study area for at least one month before the survey began. IMs are defined as individuals who do not have *hukou* in the study area, excluding individuals who are migrating for study/training purposes, tourism, and medical care.

Eight cities were included in this study, which encompassed 89 districts and counties, with varying degrees of migration and social integration programs, including Beijing (the capital of China), Zhengzhou, Chengdu (two provincial capitals), Xiamen, Qingdao, Jiaxing, Shenzhen, and Zhongshan (five economic centres in eastern and southern coastal regions). Within each city, multi-stage sampling was adopted based on the Probability Proportional to Size (PPS) method. First of all, within each district or county, subdistricts (in districts) or townships (in counties) were selected using the PPS method based on the size of the IM population in 2013. Secondly, within each subdistrict or township, neighbourhoods (in subdistricts) or villages (in townships) were selected using the PPS method based on the size of the IM population. Finally, in each village or neighbourhood, 20 IMs who met the above criteria were randomly selected. If a selected migrant was not able to be contacted or refused to participate in the study, then the next migrant listed in the sampling frame, who was the same sex and of a similar age and duration of residence in the study neighbourhood or village, was selected as the replacement. Face-to-face interviews were conducted via home visits. Interviewers from eight study cities received standardised training by the National Population and Family Planning Commission, whilst quality control was implemented during the data collection.

### 2.3. Measurement

#### 2.3.1. Outcome Measures

Mental Health: Psychological distress and perceived stress during the past 30 days were assessed. Psychological distress was measured by the 6-item Kessler Screening Scale for Psychological Distress (K6). The K6 is one of the most widely used measures of non-specific psychological distress. Its values range from 0 to 24, with higher scores indicating greater psychological distress. The K6 has been shown to have good psychometric properties amongst Chinese populations with Cronbach’s α = 0.84 [36]. During the current study, the internal consistency reliability of the K6 was 0.83.

Perceived stress was measured using the 4-item Perceived Stress Scales (PSS-4). The PSS-4 is a valid and brief measure of stress amongst a variety of the population [37,38,39,40]. The PSS-4 values range from 0 to 16, with a higher score indicating a higher level of perceived stress. Previous research confirmed the satisfactory psychometric properties of the Chinese version of PSS-4 (Cronbach’s α = 0.67) [39]. In this study, the internal reliability of the PSS-4 was 0.61.

#### 2.3.2. Independent Variable

The Internal Migrant generation’s definition was based on participants’ date of birth. According to the definition suggested by the National Bureau of Statistics of China, individuals who were born in January 1980 and later were classified as “new generation”, while those who were born before 1980 were classified as “old generation” [30]. The year 1980 was chosen because of the general social and economic reform in China that was started in that year.

#### 2.3.3. Confounding Variables

##### Migration Characteristics

Migration Characteristics include the duration of migration [14], migration path [13], and whether people are migrating with their families [40]. According to the participants’ *Hukou* status (registered place of residence is a county or a district) and current residence (a district or a county), the migration path was classified into four groups, namely rural-to-urban migration, rural-to-rural migration, urban-to-urban migration, and urban-to-rural migration.

##### Social Integration

In the current study, indicators of social integration were included as counding factors [14]. These are: The individual’s integration will (consisting of 13 questions scored on a 4-point scale, with total scores ranging from 13 to 52 and higher scores indicating a higher integration will), strong connection with place identity (e.g., thinking of oneself as native or not, using a yes/no scale), willingness to live in the current residence for the next 5 years (yes/no/not sure), views about social norms adopted (consisting of 8 questions scored on a 5-point scale, with the total scores ranging from 8 to 40, with higher scores indicating a better acculturation to local social norms), subjective income and occupation status compared with the people of the whole society (scoring from 1 to 10, with higher scores indicating a better income and occupation status), degree of respect compared with relatives, friends, and colleagues of the current residence (scoring from 1 to 10, with higher scores indicating a higher degree of respect), and type of neighbours (outsiders/locals/mixed). Questions for the social integration measurement and their rating scales are summarised in the appendix (Appendix A).

##### Demographics

To better understand the intergenerational disparity in IMs’ mental health, demographics found to be related to migrants’ mental health in previous studies were retained and included age [15,18], sex [17,18], marital status [17], education level [13], income [15,17], weekly working hours [18], and self-rated physical health [17]. Furthermore, due to the significant regional disparity in income in China [41], it is challenging to directly compare incomes across all of the 89 study districts/counties. Therefore, a ratio of the participants’ annual income to gross domestic product (GDP) per capita in 2013 in their current residence was used. Data on GDP per capita in 2013 were obtained from the study’s districts/counties

Additionally, participants’ personal autonomy was self-rated based on a total score out of 10 (1 being the worst and 10 being the best), while satisfaction with life was measured by the Satisfaction with Life scale (SWLS). The SWLS assesses global life satisfaction, with the scores ranging from 5 to 35, with higher scores indicating greater levels of life satisfaction [42]. Cronbach’s α of the SWLS was 0.88 in China [43] and 0.86 in the current study.

### 2.4. Statistical Analysis

Descriptive statistics, including the mean, standard deviation (SD), frequency, and proportion, were used to summarise demographics, migration characteristics, social integration indicators, and the mental health of the study participants. Differences between the new and old generations by study variables were assessed by the *t* test, for continuous variables, or the chi-square test, for categorical variables. When homogeneity of variance was not satisfied, Welch’s *t* test was used as an alternative to the *t* test.

The structure of our data is hierarchical; this means that IMs (level-1) were nested within study districts/counties (level-2). The intergenerational disparity in mental health was assessed using two steps. Firstly, the unadjusted differences in mental health between the two generations were assessed by two-level generalized linear mixed models (GLMMs). To build two-level GLMMs, it was assumed that mental health outcomes (the K6 score and the PSS-4 score) were distributed normally, and we only included a random intercept. Differences in mental health between new and old generations according to participants’ characteristics are summarised in Appendix A. Secondly, the association between the migrant generation and mental health was assessed via two-level GLMMs with normal distributions taking into account clustering within the district/county and control confounding variables on both levels.

Furthermore, to assess correlative factors of mental health for two IM generations separately, two GLMMs for each outcome variable were performed. The models included IMs’ demographics, migration characteristics, and social integration indicators as level-1 variables and included GDP per capita in 2013 of each district/county as a level-2 variable. Unadjusted regression coefficients (*β*), adjusted regression coefficients (*β_ad_*), and 95% confidence intervals (95% CI) were calculated. Intra-class correlation coefficients (ICC) for mental health were calculated so as to assess clustering by district/county. Analyses were conducted using the IBM SPSS Statistics 21.0 (IBM Corp. Armonk, NY, USA).

## 3. Results

### 3.1. Characteristics of Study Participants

A total of 15,999 IMs participated in this survey: 9838 (61.5%) were new generation migrants, while 6161 (38.5%) were old generation migrants. Differences in the demographics between the generations were statistically significant (*p* < 0.05). The mean age of the old and new generations was 41.8 (SD = 5.3) and 26.8 (SD = 4.5). Amongst the new generation, 46.6% were female, 40.7% were single, and 19.8% were at least college educated. By contrast, 42.5%, 4.7%, and 6.6% of old-generation migrants were female, single, and at least college educated, respectively. New-generation migrants worked an average of 53.1 (SD = 20.4) hours a week, while old-generation migrants worked an average of 57.5 (SD = 21.2) hours per week.

The migration path for new and old generants was respectively as follows: 63.7% and 60.8% for rural-to-urban migration, 23.8% and 25.6% for rural-to-rural migration, 11.2% and 11.8% for urban-to-urban migration, and 1.3% and 1.9% for urban-to-rural migration. The means (SD) of migration duration of the new and old generations were 37.0 (39.3) and 72.1 (63.8) months, respectively. Intergenerational differences in migration characteristics were significant (*p* < 0.001).

In terms of integration, the average integration will score was 39.1 (SD = 4.4) for new-generation migrants and 39.4 (SD = 4.4) for old-generation migrants (*p* < 0.001). Compared to the new generation, a greater number of the old-generation IMs indicated that they intended to live in their current residence for the next five years following the survey (67.7% vs. 53.7%, *p* < 0.001) and thought of themselves as locals (23.9% vs. 20.8%, *p* < 0.001) (Table 1).

### 3.2. The Intergenerational Disparity in the Mental Health of IMs

For new-generation migrants, the means (SD) of the K6 and PSS-4 scores were 3.5 (SD = 3.1) out of 24 and 5.4 (SD = 2.6) out of 16, respectively. The means (SD) of the K6 and PSS-4 scores among the old generation were 3.3 (SD = 3.0) and 5.2 (SD = 2.7), respectively (Figure 1). Differences between the two generations were statistically significant (*p* < 0.001 for both scales, Table 2). Descriptive statistics of the intergenerational disparity in the mental health of subgroups are summarised in the appendix Appendix A.

After controlling the participants’ characteristics and GDP per capita of each study district/county, new-generation migrants had higher psychological distress (*β_ad_* = 0.084, 95% CI: (0.026, 0.193)) and higher perceived stress (*β_ad_* = 0.118, 95% CI: (0.029, 0.207)) than the old-generation migrants (Table 2).

### 3.3. Correlative Factors of Mental Health of the New- and Old-Generation Internal Migrants

In addition, we used two-level GLMMs to examine the relationships between migration-related and individual factors and mental health by generation status(Table 3). For both generations, social integration in a variety of dimensions, self-rated physical health, personal autonomy, and life satisfaction, were negatively associated with the K6 and PSS-4 scores.

For the old-generation IMs, mental health problems were associated with being of a younger age (*β*_ad_ = −0.017, 95% CI: (−0.031, −0.002) for the K6 score) and of being female (male: *β*_ad_ = −0.180, 95% CI: (−0.303, −0.057) for the PSS-4 score). In the new generation, IMs who had a higher ratio of annual income to GDP per capita had greater psychological distress (*β*_ad_ = 0.124, 95% CI: (0.029, 0.218)), while individuals who had longer migration durations had a higher level of perceived stress (*β*_ad_ = 0.002, 95% CI: (0.000, 0.003)). In comparison with rural-to-urban migrants, urban-to-urban migrants had poorer mental health (*β*_ad_ = 0.201, 95% CI: (0.009, 0.410) for the K6, *β*_ad_ = 0.241, 95% CI: (0.073, 0.409) for the PSS-4), while rural-to-rural migrants had greater perceived stress (*β*_ad_ = 0.439, 95% CI: (0.014, 0.864)).

## 4. Discussion

Measuring the results using the K6 and the PSS-4, this study is the first of its kind to examine the psychological distress and perceived stress of two generations of IMs in China. Although they may be younger and have received a higher level of education, it was determined that new-generation IMs had a poorer mental health status than the old generation, something which is in line with evidence from the two previous studies. New-generation migrant workers were found to have worse mental health in terms of psychological symptoms and distress than the old generation [44,45]. However, the new generation’s advantage in mental health is not conclusive. The previous two studies have determined opposite outcomes in relation to the intergenerational difference in mental wellbeing amongst migrant workers [18,46]. Additionally, Yang et al. ascertained that there was no significant intergenerational disparity in depression and anxiety [47]. These inconsistent findings regarding intergenerational disparity may be due to the different measures of mental health used across all of the studies. The comparability of study results is somewhat limited. Furthermore, our previous work demonstrated that IMs in different regions of China had different age structures and migration-related characteristics, such as the province of origin, duration of migration, and the level of social integration [48]. This study established that these factors were associated with IMs’ mental health status. Therefore, variations in the migrant samples may also lead to inconclusive findings.

The intergenerational disparity in mental health could be explained by several factors. First of all, previous research has shown that young migrants were more likely to suffer from mental health problems than their predecessors [27,49,50]. However, in previous studies [30,32], significant generational differences exist, in terms of social demographics and migration-related characteristics, between the two generations. Therefore, they cannot be simply seen as two age groups of the IM population. Consequently, the age effect may only partly explain the difference. Secondly, the “salmon bias” hypothesis [11,12], which means that old migrants may have returned to their hometown after retiring or becoming ill, could potentially explain the better mental health status amongst old-generation migrants, especially if their hometowns were not part of the study areas. Thirdly, China’s social and economic reforms could also contribute to the intergenerational disparity. This study established that new-generation migrants who had a higher income, longer migration durations, and had an urban-to-urban migration background had worse mental health than their peers. It is also possible that the “one-child” policy in China has made urban-to-urban young migrants much more vulnerable than other migrants. Since the implementation of this policy in 1980, couples in urban China have only been allowed to have one child, while couples in rural areas have been allowed to have two children, so long as they met certain conditions [51]. Therefore, unlike the old generation and their rural-to-urban counterparts, urban-to-urban young migrants were more likely to be the only child in the family, which meant that they would shoulder more family expectations and responsibilities. According to Gui et al., the Chinese one-child migrant experienced an array of dilemmas from personal development in the receiving society to caring for ageing parents, due to the belief in filial piety that is deeply rooted in Chinese society [52]. Similarly, compared to rural-to-urban counterparts, urban-to-urban young migrants have a greater motivation to pursue their personal development in metropolitan areas, and the longer they have been in these areas, the less likely they are to return [13,53]. The gap between expectations and reality could also lead to intense pressure and subsequent mental health problems. Furthermore, China’s higher education reform may also be contributing to the new-generation migrants’ poorer mental health. For example, China embarked on a higher education reform to increase college enrolment in 1998, when the first batch of new generation migrants were 18 years old [54]. The expansion of higher education has trained and produced more young migrant graduates than there are decent jobs available. The limited number of employment opportunities post-graduation means that new-generation IMs now face upward social mobility in cities and challenging working environments as sources of pressure. The mental health consequences of unemployment are well documented in the literature [55,56].

The findings of this study underscore the importance of targeting young migrants in providing mental health services. Future research should pay particular attention to the needs and trends of new-generation migrants, especially urban-to-urban migrants and migrants who have high-paying jobs. They are traditionally considered to have advantages and, as a result of this, tend to be understudied. Furthermore, social integration services, such as providing education in a socio-cultural context, along with providing and encouraging migrants to participate in social activities, may improve IMs’ mental health.

Our findings are not able to answer the research question about whether migrants are more likely to suffer from distress/psychological distress than their counterparts in a non-IM generational cohort. While it is not an ideal comparator, our study does suggest that the K6 and PSS-4 scores amongst IMs were lower than they are in existing data reported in other Chinese cohorts, such as undergraduate students and cardiac patients [36,39].

This study does have a few limitations. First of all, the “salmon bias” [11] may cause an underestimation of the old-generation migrants’ mental health. The salmon bias hypothesis has been established amongst returned migrant workers in relation to self-rated physical health but not psychological health [57]. More in-depth research about these returning migrants’ mental health is needed. Secondly, the applicability of mental health models for the old generation, especially less educated IMs, should be assessed in the future. In mainland China, the idea of mental health is more or less a recently developed academic and scientific topic. Thus, those who are from an older generation may be less likely to be familiar with expressing themselves in the same idioms of mental health as younger generations. Thirdly, this sample study is limited to migrants between the ages of 18 and 59. Therefore, our findings cannot make generalizations about IMs in all age groups. Finally, causation cannot be defined due to the cross-sectional nature of the study. Future studies, using prospective designs, are needed to provide additional support for predictors of IMs’ mental health.

Despite these limitations, this study has assessed the intergenerational difference in IMs’ mental health, which, to some extent, reflects how social and economic reforms have affected mental health in China. Compared to migrants who are born after China’s economic reform and the adoption of the one-child policy, old-generation IMs have a small but significant advantage when it comes to mental health. These findings suggest that efforts to promote the mental health of IMs may be more useful if they are targeted at young migrants, especially migrants who earn a higher income and those migrants who have an urban-to-urban migration background. Services for addressing generation-specific needs, for example, social integration services, may help facilitate mental health promotion amongst IMs in China.

## Figures and Tables

**Figure 1 ijerph-16-02608-f001:**
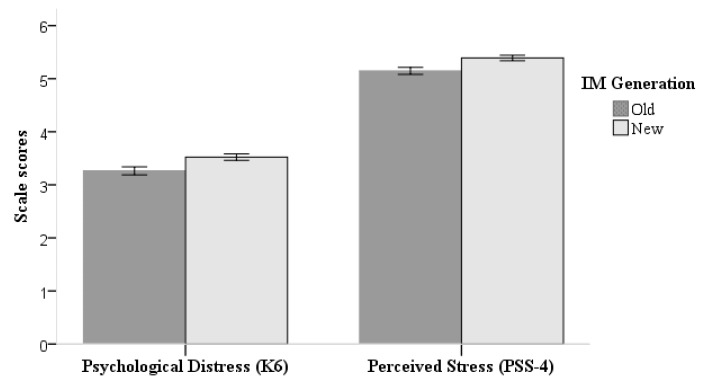
Differences in mental health between new- and old-generation migrants. Abbreviation: IM = internal migrant. Error bars represent 95% confidence intervals.

**Table 1 ijerph-16-02608-t001:** Characteristics of 15,999 internal migrants (IMs) in China, 2014.

Characteristics	Generation ^a^		Total ^a^ (*n* = 15,999)
	New (*n* = 9838)	Old (*n* = 6161)	
Demographics			
Age mean (SD)	26.8 (4.5)	41.8 (5.3)	32.6 (8.7) ***
Weekly working hours mean (SD)	53.1 (19.7)	57.7 (21.2)	54.8 (20.4) ***
Annual income/Regional GDP per capita 2013 mean (SD)	0.65 (0.67)	0.68 (0.87)	0.66 (0.76) *
Sex *n* (%)			
Male	5258 (53.4)	3541 (57.5)	8799 (55.0) ***
Female	4580 (46.6)	2620 (42.5)	7200 (45.0)
Marital status *n* (%)			
Married	5835 (59.3)	5874 (95.3)	11,709 (73.2) ***
Single	4003 (40.7)	287 (4.7)	4290 (26.8)
Education level *n* (%)			
Primary school and less	307 (3.1)	1198 (19.4)	1505 (9.4) ***
Secondary school	4687 (47.6)	3398 (55.2)	8085 (50.5)
High school	2892 (29.4)	1159 (18.8)	4051 (25.3)
College and above	1952 (19.8)	406 (6.6)	2358 (14.7)
Self-rated physical health *n* (%)			
Good	8992 (91.4)	5210 (84.6)	14,202 (88.8) ***
Fair	827 (8.4)	911 (14.8)	1738 (10.9)
Poor	19 (0.2)	40 (0.6)	59 (0.4)
Migration Characteristics			
Duration of migration (months) mean (SD)	37.0 (39.3)	72.1 (63.8)	50.5(53.0) ***
Migration path *n* (%)			
Rural–Urban	6267 (63.7)	3745 (60.8)	10,012 (62.6) ***
Urban–Urban	1102 (11.2)	725 (11.8)	1827 (11.4)
Rural–Rural	2340 (23.8)	1575 (25.6)	3915 (24.5)
Urban–Rural	129 (1.3)	116 (1.9)	245 (1.5)
Migrating with families *n* (%)			
No	3534 (35.9)	147 (2.4)	3681 (23.0) ***
Yes	6304 (64.1)	6014 (97.6)	12,318 (77.0)
Social Integration			
Integration will (13–52) mean (SD)	39.1 (4.4)	39.4 (4.4)	39.2 (4.4) ***
Views about social norms adopted (8–40) mean (SD)	24.1 (4.1)	23.4 (4.1)	23.8 (4.1) ***
Income; occupation position compared with the people of the whole society (0–10) mean (SD)	4.6 (1.7)	4.7 (1.7)	4.6 (1.7) **
Degree of respect compared with relatives, friends and colleagues of the current residence (0–10) mean (SD)	6.0 (1.6)	6.0 (1.6)	6.0 (1.6)
Willingness to live in current residence for the next 5 years *n* (%)			
Yes	5287 (53.7)	4169 (67.7)	9456 (59.1) ***
No	1231 (12.5)	571 (9.3)	1802 (11.3)
Not sure	3320 (33.7)	1421 (23.1)	4741 (29.6)
Type of neighbours *n* (%)			
Outsiders	4389 (44.6)	2564 (41.6)	6953 (43.5) ***
The locals	1959 (19.9)	1344 (21.8)	3303 (20.6)
Mixed	3489 (35.5)	2252 (36.6)	5741 (35.9)
Thinking oneself native or not *n* (%)			
Yes	2042 (20.8)	1474 (23.9)	3516 (22.0) ***
No	7795 (79.2)	4686 (76.1)	12481 (78.0)
Personal Autonomy (0–10) mean (SD)	6.7 (1.8)	6.8 (1.8)	6.7 (1.8) ***
Satisfaction with Life (5–35) mean (SD)	21.4 (6.2)	22.6 (6.3)	21.9 (6.2) ***

Abbreviations: SD = Standard Deviation, GDP = Gross Domestic Product. *: *p* < 0.05; **: *p* < 0.01; ***: *p* < 0.001. ^a^: Numbers may not add to column total, due to missing data.

**Table 2 ijerph-16-02608-t002:** The intergenerational disparity in the mental health of internal migrants, 2014.

Characteristics	Psychological Distress		Perceived Stress	
	*β* (95% CI)	*β_ad_* (95% CI) ^#^	*β* (95% CI)	*β_ad_* (95% CI) ^#^
**Generation**				
Old (Ref.)	0	0	0	0
New	0.197 (0.099, 0.295) ***	0.084 (0.026, 0.193) *	0.222 (0.139, 0.305) ***	0.118 (0.029, 0.207) **
**Variance (estimates)**				
*Level-2*	0.814		0.953	
*Level-1*	8.952		6.435	
**ICC (%)**	8.34		12.90	

Abbreviations: Ref. = Reference Group; ICC = Interclass Correlation Coefficient; *β* = unadjusted regression coefficient; *β_ad_* = adjusted regression coefficient. *: *p* < 0.05; **: *p* < 0.01; ***: *p* < 0.001. **^#^** Adjusted for participants’ demographics, migration characteristics, and social integration indicators (level-1 variables), and GDP per capita in 2013 of each district/county (level-2 variable).

**Table 3 ijerph-16-02608-t003:** Correlative factors of mental health of new- and old-generation internal migrants, 2014.

Characteristics	Psychological Distress *β_ad_* (95% CI) ^#^		Perceived Stress *β_ad_* (95% CI) ^#^	
	New Generation	Old Generation	New Generation	Old Generation
**County-level**				
GDP per capita	−0.007 (−0.045, 0.030)	0.022 (−0.020, 0.064)	−0.013 (−0.052, 0.025)	0.025 (−0.011, 0.061)
**Individual-level**				
**Demographics**				
Age	0.001 (−0.016, 0.019)	−0.017 (−0.031, −0.002) *	−0.011 (−0.025, 0.004)	−0.010 (−0.022, 0.002)
Weekly working hours	−0.004 (−0.008, 0.001)	0.002 (−0.002, 0.006)	0.001 (−0.003, 0.004)	−0.002 (−0.006, 0.002)
Annual income/Regional GDP per capita 2013	0.124 (0.029, 0.218) *	−0.037 (−0.124, 0.049)	0.057 (−0.192, 0.305)	0.023 (−0.164, 0.210)
Sex				
Female (Ref.)	0	0	0	0
Male	−0.015 (−0.133, 0.102)	−0.114 (−0262, 0.034)	−0.020 (−0.114, 0.075)	−0.180 (−0.303, −0.057)**
Marital status				
Single (Ref.)	0	0	0	0
Married	−0.001 (−0.312, 0.309)	0.159 (−0.316, 0.634)	0.057 (−0.192, 0.305)	−0.026 (−0.421, 0.369)
Education level				
College and above (Ref.)	0	0	0	0
Primary school and less	−0.122 (−0.049, 0.248)	−0.234 (−0.610, 0.141)	0.265 (−0.033, 0.562)	−0.262 (−0.576, 0.052)
Secondary school	−0.275 (−0.456, −0.094) **	−0.294 (−0.627, 0.038)	0.049 (−0.096, 0.193)	−0.262 (−0.541, 0.016)
High school	−0.083 (−0.262, 0.096)	0.043 (−0.294, 0.381)	0.089 (−0.054, 0.233)	−0.266 (−0.548, 0.016)
Self-rated physical health				
Poor (Ref.)	0	0	0	0
Good	−1.944 (−3.359, −0.528) **	−1.700 (−2.727, −0.673) **	−1.811 (−2.946, −0.676) **	−0.923 (−1.799, −0.067) *
Fair	−0.760 (−2.188, 0.668)	−1.049 (−2.087, −0.011) *	−0.984 (−2.129, 0.161)	−0.423 (−1.3289, 0.442)
**Migration Characteristics**				
Duration of migration (months)	−0.000 (−0.002, 0.002)	0.000 (−0.001, 0.002)	0.002 (0.000, 0.003) *	−0.000 (−0.001, 0.001)
Migration path				
Rural–Urban (Ref.)	0	0	0	0
Urban–Urban	0.201 (0.009, 0.410) *	0.097 (−0.160, 0.354)	0.241 (0.073, 0.409) **	0.127 (−0.087, 0.341)
Urban–Rural	0.162(−0.013, 0.337)	0.209 (−0.006, 0.424)	0.046 (−0.098, 0.189)	0.162 (−0.018, 0.342)
Rural–Rural	0.410 (−0.118, 0.939)	0.054 (−0.485, 0.593)	0.439 (0.014, 0.864) *	0.098 (−0.352, 0.548)
Migrating with families				
Yes (Ref.)	0	0	0	0
No	0.140 (−0.161, 0.440)	0.394 (−0.252, 1.039)	0.144 (−0.097, 0.384)	0.447 (−0.091, 0.984)
**Social Integration**				
Integration will	−0.040 (−0.054, −0.026) ***	−0.029 (−0.046, −0.012) **	−0.016 (−0.028, −0.005) **	−0.027 (−0.041, −0.013) ***
Views about social norms adopted	−0.070 (−0.086, −0.055) ***	−0.069 (−0.088, −0.051) ***	−0.042 (−0.054, −0.030) ***	−0.042 (−0.058, −0.027) ***
Income, occupation position compared with the people of the whole society	−0.044 (−0.084, −0.004) *	0.006 (−0.044, 0.056)	0.003 (−0.029, 0.035)	0.04 7(0.006, 0.089) *
Degree of respect compared with relatives, friends, and colleagues of the current residence	−0.018 (−0.060, 0.024)	−0.007 (−0.058, 0.044)	−0.066 (−0.100, −0.033) ***	−0.092 (−0.134, −0.049) ***
Willingness to live in current residence for the next five years				
No (Ref.)	0	0	0	0
Yes	0.087 (−0.105, 0.278)	−0.142 (−0.396, 0.113)	−0.201 (−0.354, −0.048) *	−0.264 (−0.477, −0.052) *
Not sure	−0.119 (−0310, 0.072)	−0.014 (−0.289, 0.262)	−0.097 (−0.252, 0.056)	0.010 (−0.219, 0.240)
Type of neighbours				
Outsiders (Ref.)	0	0	0	0
The locals	0.009 (−0.165, 0.183)	0.043 (−0.164, 0.249)	−0.062 (−0.202, 0.078)	0.004 (−0.168, 0.177)
Mixed	−0.104 (−0.242, 0.034)	−0.075 (−0.242, 0.093)	0.030 (−0.081, 0.141)	−0.029 (−0.168, 0.110)
Thinking oneself native or not				
No (Ref.)	0	0	0	0
Yes	0.036 (−0.117, 0.188)	0.139 (−0.040, 0.318)	−0.125 (−0.247, −0.002) *	−0.084 (−0.233, 0.065)
**Personal autonomy**	−0.403 (−0.441, −0.364) ***	−0.385 (−0.432, −0.339) ***	−0.405 (−0.436, −0.374) ***	−0.397 (−0.435, −0.358) ***
**Satisfaction with life**	−0.070 (−0.081, −0.060) ***	−0.078 (−0.091, −0.064) ***	−0.088 (−0.096, −0.079) ***	−0.096 (−0.107, −0.085) ***
***Variance (estimates)***				
*Level-2*	0.745	0.790	0.896	0.907
*Level-1*	9.113	8.620	6.317	6.575
***ICC* (%)**	7.56	8.40	12.42	12.12

Abbreviations: GDP = Gross Domestic Product; Ref. = Reference Group; ICC = Interclass Correlation Coefficient; *β_ad_* = adjusted regression coefficient. *: *p* < 0.05; **: *p* < 0.01; ***: *p* < 0.001.

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
