# Peer review of "The Disparity in Mental Health Between Two Generations of Internal Migrants (IMs) in China: Evidence from A Nationwide Cross-Sectional Study"

_ijerph, 2019, doi:10.3390/ijerph16142608_

Round 1
Reviewer 1 Report
The disparity in mental health between two
generations’ internal migrants in China:evidence
from a nationwide cross-sectional study
This is an interesting study on mental health differences between two generations of internal migrants in China and its correlates. Most research on the relationship between migration and mental health is on international migrants. So this study may have an important contribution to this field, also because of the link with the social-economic policy in China.
A few improvements in the text are needed, but the way the analysis is clarified and presented needs major revision. Please, find my suggestions for improvement below.
ABSTRACT
‘… Existing evidence suggests disparities in mental health
between IMs who born before and after 1980, when the general social and economic reform started
in China is not clear. …’
Please, reconsider this sentence. It is not clear what is not clear.
‘… For new generation IMs, poor mental
health was associated with urban-to-urban migration path (β=0.060, 95%CI: 0.010-0.111 for the K6,
β=0.241, 95%CI: 0.073-0.409 for the PSS-4), migration duration (β=0.002, 95%CI: 0.000-0.003 for the
PSS-4), and annual income (β=0.052, 95%CI: 0.003-0.100 for the K6. …’
Indicate the direction of the found associations: that is: longer migration duration means worse mental health? And higher income better mental health?
1. INTRODUCTION
‘… 1.2 The intergenerational disparity in migrants’ health …’
Second generation immigrants means that the parents were born in the country of origin but their child/ children in the country of destination. That is different from the meaning of the word ‘generation’ in the present paper, where ‘younger generation’ means being born after 1980. I wonder whether this paragraph is relevant for your study.
Differences in mental health between those born before and in 1980 and thereafter has probably to do with differences in China’s migration policy and associated selection of person’s characteristics but not with differences in country of birth.
‘… Available evidence suggests these differences may lead to differences in mental health between generations but with little empirical evidence [30-32]. …’
It is not clear why female gender, higher educational attainment and more diverse networks imply higher risk of mental health problems.
2. METHODS
‘… For either new or old generation, correlates of IMs’ mental health were assessed by two-level GLMMs with normal distributions. …’
Please, explain what is assumed to be normally distributed? The mental health scales? So there are different models, each with a different mental health outcome (K6, PSS-4, SWLS)?
The outcome should be normally distributed, but the predictors not necessarily. Why use a mixed model? To take into account the clustering within district/ counties. Adjustment for confounding does not make use of a multilevel model essential.
3. RESULTS
Table 1 Please, explain the meaning of * and ** when *** means P value<0.001.< span="">
There are some minor differences that are still statistically significant.
Eg:
Annual income/Regional GDP per capita 2013 mean (SD) 0.7(0.7) 0.7(0.9) 0.7(0.8)*
What is the difference in GDP actually?, and does the P value make any sense?.
There are some differences in the SD between the two groups, eg for the variable age (4.5 vs 5.3). Does the used T test take that into account, eg by using the Welch variant, or was homogeneity in variance assumed?
Table 2 what do the figures after Age r, Weekly working hours mean?
Does a higher age mean a lower Psychological distress among New generation (-0.03) and old generation (-0.04)?
And what does the P value of *** mean: that there is a difference in estimated r between the generations, or that in general (Total sample) the r coefficient is significantly different from 0.0?
What is the use of all these detailed figures?
What is important is whether the difference in mental health between young and old generation is still present after adjustment, and if not, what factor(s) is (are) influential in explaining the intergenerational difference.
Consider to put this Table in supplement and give a short summary of the relevance for the next section of the analysis (Table 3).
‘… After controlling for a range of participants’ characteristics, new generation migrants had higher
psychological distress (β=0.022, 95%CI: 0.005-0.049), and higher perceived stress (β=0.118, 95%CI:
0.029-0.207) than old generation migrants (Table 3). ...’
Were the given estimates for the effect of generation adjusted for all factors shown in Table 3?
Were factors present that explained a part of the differences in mental health between the generations?
It may be informative to give an unadjusted effect estimate from GLMM model and one adjusted for the other factors.
Table 3 presentation is confusing. The relevance of all these detailed figures is hard to grasp.
For both new and old generation there are different estimates for the effect of GDP on psychological distress. Does that mean that terms for {GDP x generation} interaction were included? But that does necessarily imply that that the effect of generation on psychological distress was also different for different values of GDP. However, there was only one effect estimate of generation on mental health (separately for K6 and PSS-4) estimated. And this effect was put in column ‘Overall’.
Now generation is defined as >= 35 years old vs <= 34 years old. Please, use the definion given in the Methods for clarity (albeit both will match in a cross sectional design).
‘… We further used two-level GLMMs to examine correlates of two migrant generations’ mental
3 health separately. The results are summarised in Table 3. …’
Does that mean that there are two analyses presented in one Table? Both the analysis with adjustment for confounding and one effect of generation on mental health, and the analysis with estimates separately for young and old generation? Please, explain,and present two different analyses in two different tables.
4. DISCUSSION
‘… Our study found that compared to the old generation, new generation migrants’ mental
health was more likely to be associated with demographical and migration characteristics, with new
generation migrants with higher income, longer migration durations and with urban-to-urban
migration background having worse mental health. …’
But Table 1 shows much shorter migration duration among younger generation, as expected.
This sentence may be misunderstood, as it is about the associations within the two generations, not the difference between the generations.
‘… Future research should pay attention to needs and trends of new generation
migrants, especially new migrants, are traditionally considered to have more advantages and are
understudied, including urban-to-urban migrants and migrants with high paying jobs. …’
The flow of this sentence fails a little.
Reviewer 2 Report
The paper is of interest of wider academic and worth publishing. However. As it stands now, I do not feel it is ready for publication. In general, the literature review is a bit “biased” and does not include more recent evidence. Some suggestions have been done.
More explanations are needed overall, and the main results (Table 3) are not clearly presented and not enough discussed. Below some detailed comments.
Main issues:
1. The findings that New generation 36 migrants had worse mental health compared to older IMs is in contrast with existing literature. However, this should be discussed and also the reasons for the difference with existing literature, as well as the reasons why young migrants have a worse mental health should be discussed.
2. Page 2: it is unclear what happened in 1980, the authors say:” However, in the 1980s, China instigated market-oriented reforms to boost its economy.” But this does not explain why people started to migrate. This needs to be clarified.
3. Literature review is not necessarily up to date, and some citations are incomplete or incorrect: example includes:
4. “Previous research on migration and health was mainly conducted among international migrants 11 from developing countries to developed countries”, this is not necessarily true: a larger share of migrants in the UK are, for example, from developed countries such as Germany, France, Italy, Poland, Ireland, US and so on.
5. “It has 16 been hypothesised that migration in itself and most jobs for migrants in receiving countries are 17 physically demanding”. This is not necessarily the case for the US and the UK where while it tis true many migrants work in physically demanding job, a high percentage ( about 50%) have high level of education and are employed in high skill jobs. See the briefing from the Migration Observatory, for more details.
6. “International studies have demonstrated intergenerational disparity in migrants’ health, and the 42 young generation appeared to have poorer health.” Again, this is not necessarily true, see recent evidence from “Intergenerational and inter-ethnic mental health: an analysis for the UK" (with R. Dorsett; Rienzo, C. and M. Weale) 2018 Population, Space and Place, and Vargas-Silva, C., Giuntella, O., Mazzonna, F., Nicodemo, C. et al. (2018) Immigration and the reallocation of work health risks, Journal of Population Economics, pp 1 – 34.
7. The way Table 3 is presented is confusing. Unclear what the 0 is in the correlates. The presentation should and could be improved. Too much stuff and no much interpretation or explanation. Also, why not report only the standard error?
8. Too many variables presented at the same time.
Minor issues:
1. English could be improved: examples include a) abstract sentence not grammatically clear/correct:
2. “Existing evidence suggests disparities in mental health between IMs who born before and after 1980, when the general social and economic reform started in China is not clear”
3. Page 3: sentence “born in 1980 3 or later when China started economic reform and adopted the “one-child” policy” how does the one-child policy interacts with migration? This deserves some explanations.
4. I would replace “participants’ birthday” with “participants’ date of birth”.
5. Page 4: till now it is still unclear the cut off of new and old generation before and after 1980. I do not feel this has been explained well or sufficiently.
6. Differences between new and old generations by study 40 variables were assessed by the t test: this is repeated few lines later.
7. Tables: tables would be more easy to read if variables were aligned to the left.
Reviewer 3 Report
This manuscript addresses an interesting and important topic about mental health among internal migrants in China, but there are a number of weaknesses to the current study.
First, it is questionable whether and to what extent literature on international migration is useful for understanding internal migration, as there are fundamentally different sociocultural, linguistic, political, and geographical factors involved. These differences need to be critically engaged with early on in the manuscript.
Second, although the study purports to look at different generations of IMs, and references studies comparing different generations of international migrants in the U.S. and Europe, I believe that the definition of generations in the U.S. and Europe usually refer to first-generation, second-generation, etc. in terms of whether they moved to the receiving country (first-generation), or were born in the receiving country to migrant parents (second-generation and beyond). In contrast, the present study seems to look at just the age of migrants as the definition of generation, rather than whether or not they moved to or were born in the receiving area. These differences in definitions of generation as well as their analytical import need to be more clearly explained.
Third, the study found comparatively worse mental health conditions among the younger generation of IMs, but this should be put into context with their non-IM generational cohort to determine to what degree internal migration is related to their mental health, and to what degree their mental health conditions are a shared aspect of their generational cohort, regardless of IM status. In addition, the applicability of certain mental health models for older generations must also be assessed, especially since the idea of mental health as an academic and scientific object of study and concern is more-or-less a recent development in mainland China, and thus older (especially less educated) individuals are less likely to be familiar with expressing themselves in the same idioms of mental health/distress as younger generations.
Altogether, there is significant potential in this study, but the various definitions and limitations need to be more carefully considered. Thorough English editing is also necessary.
Round 2
Reviewer 1 Report
The authors did a lot of work and the paper has been significantly improved.
After some minor revision the paper warrants publication in IJERPH. See my few comments for final revision below.
Methods
Page 5, line 10 ‘…The structure of our data is hierarchical; this means that IMs (level‐1) were nested within study
districts/counties (level‐2)….’
>> it is better explained now. Please, clarify whether both a random intercept and random slope were included in the modelling, or only random intercept or the correlation between observations was modelled in a different way.
Figures in Table suggest that there is only a random intecept. Is that right?
Level-1(σ2u0) 0.814 0.953
Level-2(σ2e) 8.952 6.435
ICC (%)
I wonder whether σ2u0 is related to random intercept, and thus to level 2, not level 1.
And σ2e to the variance at individual data-point level, thus to level 1.
>> Page 5, line 14 ‘…In addition to this, we analysed
differences in mental health between new and old generations in accordance with participants’
characteristics
in accordance with … you probably mean …according to… is that right?
Reviewer 2 Report
Dear Authors,
many thanks for your time and effort to address my concerns. I am pleased to see you have considered them all.
I fell the paper is much more improved now and ready for publication.
I must say I still disagree with some of the aspects reported for the literature such as the sentence: "Previous research findings on migration and health among international migrants have postulated the 18 ‘healthy migrant paradox’, which stipulates that recent migrants from developing countries appear to have 19 better overall health, especially physical health, than permanent residents in host countries and those who 20 remain behind in their own countries; unfortunately this health advantage dissipates over time [5‐8]. It has 21 been hypothesised that migration itself is physically demanding and migrants tend to self‐select jobs which 22 have a high physical burden [9]. The pre‐migration self‐selection process and health screening, cultural 23 buffering and employability of these receiving countries cannot solely explain this epidemiological paradox, 24 whereby the healthy migrant effect is clearly prevalent despite the fact that these migrants are coming from 25 poor countries that have a lower socioeconomic status and do not have as much access to healthcare than 26 the receiving population [7, 8]. "
But i understand your position. Also, in the comments you refer to the paper of Vargas-Silva, C., Giuntella, O., Mazzonna, F., Nicodemo, C. et al. (2018) Immigration and the reallocation of work health risks, Journal of Population Economics, pp 1 – 34 but you do not cite the paper.
Thanks, at the end it is indeed a nice paper.
Reviewer 3 Report
The authors have shown significant effort in revising the manuscript and responding the various concerns of the reviewers. The authors acknowledge some of the limitations of their study, which I still feel are important to be addressed, but I understand that it is not possible with the current data, and thus I accept that the data presented here will be a useful resource for future research.
Apart from some minor language editing, I believe that the manuscript is ready to be published.
